# Determinants of patient satisfaction: Lessons from large-scale inpatient interviews in Vietnam

**Thuy Nguyen**[1]*, **Huong Nguyen**[2], **Anh Dang**[3]

**1** School of Public Health, University of Michigan, Ann Arbor, Michigan, United States of America, **2** O'Neill School of Public and Environmental Affairs, Indiana University, Bloomington, Indiana, United States of America, **3** National Center for Socio-Economic Information and Forecast, Ministry of Planning and Investment of Vietnam, Hanoi, Vietnam

* thuydn@umich.edu

**Data Availability Statement:** The minimal anonymized data set and Stata codes necessary to replicate our study findings is available on Github, a public repository (https://github.com/nguyendieuthuy/PatientSatisfaction.git). The full

## Abstract

Patient satisfaction, a healthcare recipient's reaction to salient aspects of their service experience, has been considered an important metric of the overall quality of healthcare in both advanced and developing countries. Given the growing number of studies on patient satisfaction in developing and transitioning countries, publications using high-quality patient surveys in these countries remain scarce. This study examines factors associated with inpatient satisfaction levels using nationwide, large-scale interview data from 10,143 randomized and voluntary patients of 69 large and public hospitals in Vietnam during 2017-2018. We find that older patients, poor patients, female patients, patients with lower levels of education, patients not working for private enterprises (or foreign enterprises), and rural patients reported higher levels of overall satisfaction. Health insurance is found to have positive influence on overall patient satisfaction, primarily driven by limiting patient concerns about treatment costs, as well as increasing positive perceptions of hospital staff. We further find that patients who paid extra fees for their hospital admission expressed higher scores for hospital living arrangements and medical staff, but were mostly dissatisfied with treatment costs. These findings have important policy implications for current policy makers in Vietnam as well as for other countries with limited healthcare resources and ongoing healthcare reforms.

## Introduction

Patient satisfaction is defined as "a healthcare recipient's reaction to salient aspects of the context, process, and result of their service experience" [1]. In order to measure patient satisfaction, it is necessary to evaluate patient perceptions and to determine whether or not the patients perceived their medical needs were adequately met [2]. In advanced countries, patient-oriented outcomes and patient satisfaction surveys are central to designing and evaluating healthcare services and delivery system by reflecting the quality of services from the patients' perspective and identifying patients who need additional attention [3]. For example,

dataset is available upon request to (Dr. Huong Nguyen, e-mail: huong.lan. nguyen@sangkienvietnam.org).

**Funding:** TN and HN received no specific funding for this work. Anh Dang acknowledges funding from National Foundation for Science and Technology Development (NAFOSTED), Vietnam under grant number 502.01-2015 Funding information: National Foundation for Science and Technology Development (NAFOSTED), Vietnam under grant number 502.01-2015 Dr. Anh Dang" https://nafosted.gov.vn/en.

**Competing interests:** The authors have declared that no competing interests exist.

evaluation of patient satisfaction has been required in all French hospitals since 1996 and in Germany since 2005 [4]. In developing countries, recording patient views on healthcare delivery is becoming increasingly important as these countries shift from a doctor-to-patient relationship to the more modern provider-client attitude [5].

This study examines factors associated with inpatient satisfaction levels in Vietnam's public hospitals during 2017-2018, using nationwide, large-scale interview data from randomized and voluntary responses to a survey provided to patients of 69 large and public hospitals. We look at how patient satisfaction levels respond to service-specific characteristics (length of stay and extra-paid services), patients' financial conditions (employment status, health insurance, and income level), and demographic factors (age, gender, race, location and level of education). We further evaluate the associations of these factors with different salient aspects of their healthcare service experience: hospital staff, hospital facilities, and treatment costs.

The data show that elder patients, low-income patients, female patients, those with only primary level education, and rural patients tended to report higher scores on overall satisfaction of public hospitals' services in Vietnam. More importantly, health insurance is positively associated with a higher level of patient satisfaction which may be primarily driven by reducing patient concerns about treatment costs. Patients who paid extra fees for their hospital admission reported higher scores for hospital living arrangements and medical staff, but were mostly dissatisfied with treatment costs.

This study attempts to close the existing gap in prior work on understanding patient satisfaction in Vietnam which is limited in specific health clinics and presents a major geographic limitation. Although studies on patient satisfaction have been increasing in developing countries including Vietnam, publications using high-quality and large-scale patient surveys are still scarce due to resource and time constraints. Despite a number of studies which have been conducted in Vietnam to evaluate patient satisfaction in a general inpatient setting or in specific healthcare clinics, these studies often present a major geographic limitation. For instance, one study explored psychological and socioeconomic factors associated with patients' evaluation of healthcare quality, using data from over 2000 patients in a survey conducted in Hanoi in 2016 [6]. They found that the average score on health services quality was 3.55 (where 5 is the highest score), which was positively associated with younger age, higher-levels of income, and better health conditions. Three other studies were conducted only in the North of Vietnam [7–9]. Nguyen and Nguyen (2014) conducted a patient survey on 894 patients in 18 public hospitals in the North of Vietnam and suggested the average satisfaction score as 3.68 based on a 5-point Likert scale [7]. Vuong et al. (2018) also conducted a survey comprising of 900 patients in five provinces in Northern Vietnam from August 2014 to June 2015 [9]. The authors found a low satisfaction level from this group of patients: about 66% of participants perceived unsatisfactory levels about the health services. Other studies addressing some extent of geographical differences, however, were limited to some specific clinic setting [10, 11]. Tran et al. evaluated patient satisfaction with HIV care in Hanoi, Hai Phong (two provinces in Northern Vietnam), and Ho Chi Minh City (in Southern Vietnam) [10]. Interviewing 1016 patients in these clinics, the authors found that the percentage of respondents completely satisfied with overall service quality and treatment outcomes was 42.4% and 18.8%, respectively, which was considerably lower than the satisfaction level in general healthcare settings. To the best of our knowledge, our paper is the first study on patient satisfaction in Vietnam which addresses these geographic and specific clinic setting limitations by using a nationwide and large-scale interview data.

This paper contributes to the medical literature regarding determinants of patient satisfaction by providing analyses using a recent sample of data on 10,143 randomized and voluntary patients in Vietnam. Examining the correlation between demographic factors, health status,

and multidimensional attributes of healthcare settings with patient satisfaction, prior studies provide mixed findings [4]. In a recent systematic review, Batbaatar et al. (2017) identified a total of 71 studies between 1980 and 2014 which often provided week and contradictory findings across samples on the relationship between various patient-related characteristics and patient satisfaction [12]. For instance, higher income group patients tended to be more satisfied with overall health services in a number of studies while another study concluded that lower-income patients were more satisfied with nursing care [12]. In addition, one study in Scotland found that patients with only primary education level reported higher patient satisfaction using data of 650 patients discharged from four acute care general hospitals during February and March 2002 [13]. One similar study showed that education level was not a significant predictor of overall patient satisfaction using a Norwegian national patient-experience survey of 10,912 patient [14]. These examples suggest that it is important to obtain more representative samples to attain better understanding of patient satisfaction.

Analyzing various demographic and economic characteristics of patients and services may help us better understand the quality of hospital services from the patients' perspective and identify which sub-populations need additional attention. The focus of this study is Vietnam, the world's fifteenth most populous country with a rapidly developing economy. The current healthcare system in Vietnam is a mixed public-private provider system under a number of initiatives and health finance reforms, which aims toward universal health coverage and improved general public health [15]. These findings have important policy implications for policy makers in Vietnam as well as for other countries with limited healthcare resources and ongoing healthcare reforms.

## Materials and methods

### Design and setting

This cross-sectional patient-level study is a secondary data analysis using a nationwide and large-scale interview data. In particular, this study used 10,143 randomized and voluntary responses of participants from 69 large and public hospitals located in 27 Northern, Central, and Southern provinces and cities in Vietnam in 2017-2018. Eligible participants were inpatients (or close relatives who accompanied patients during the treatment process and answered on behalf of the patients) satisfying: (1) at least 18 years old, (2) stayed at least one entire day in one of participated hospitals from April to July in the surveyed year, (3) survived after treatment.

The participating hospitals are located nationwide in 27 provinces and cities of Vietnam. The locations of 29 participating hospitals in 2017 include: (1) 10 Northern provinces including Ha Giang, Son La, Bac Giang, Bac Ninh, Hanoi, Hung Yen, Nam Dinh, Thai Binh, and Ninh Binh; (2) 7 Central provinces including Nghe An, Quang Tri, Da Nang, Quang Ngai, Hue, Ninh Thuan, and Binh Dinh; (3) 4 Southern provinces including Ho Chi Minh City, Can Tho, Tien Giang, Ba Ria Vung Tau, and Ca Mau. The locations of the 60 participating hospitals in 2018 include: (1) 8 Northern provinces including Ha Giang, Son La, Bac Giang, Thai Nguyen, Quang Ninh, Hanoi, Ninh Binh, and Thanh Hoa; (2) 8 Central provinces including Nghe An, Quang Tri, Da Nang, Quang Ngai, Hue, Ninh Thuan, Binh Dinh, and Khanh Hoa; and (3) 7 Southern provinces including Ho Chi Minh City, Can Tho, Tien Giang, Ba Ria Vung Tau, Ca Mau, Vinh Long, and Hau Giang. Among these 60 participating hospitals in the 2018 survey, 21 hospitals were participated in the 2018 survey. The health system of Vietnam consists of the central level, provincial level, district level, and commune level. The surveys in this study covered three levels of hospitals in Vietnam: central, provincial, and

district [16]. The lowest level of Vietnam's healthcare system, commune health stations were not included in these surveys.

This secondary data analysis was determined to be exempt from review and informed consent by the institutional review board of the Indiana University Human Subject Office because the interviews were implemented by Vietnam Initiative for non-research purposes prior to this study. In addition, data of this paper were obtained from Vietnam Initiative in a fully anonymized and de-identified manner.

## Data source

Data of this study comes from Vietnam Initiative. This institution designed the interview methods and contracted an independent interviewing organization to conduct these interviews for the project titled "Equitable Healthcare through Patient Satisfaction Index". Fig 1 provides detailed steps of the sampling approach and data collection process implemented by Vietnam Initiative. In step 1, in collaboration with Vietnam Initiative, Medical Services Administration (MSA, operating under the Ministry of Health) of Vietnam sent the official letter to 80 public hospitals in 2018 and 50 hospitals in 2017. The letter called for voluntary participation of hospitals without any financial incentives. In step 2, the participating hospitals used the secured link provided in the letter to supply the lists of all eligible patients with cellphone numbers to Vietnam Initiative. For example, Vietnam Initiative received a list of 136,678 inpatients who stayed in one of the 29 participated public hospitals from April to July 2017.

In step 3, Vietnam Initiative de-identified patient lists and randomly selected up to 250 patients from each hospital in 2017 (or up to 500 patients in 2018). For five small hospitals with the number eligible patients under 500, all eligible patients were included. The de-identified lists were sent to an independent survey contracter, the Development and Policies Research Center (DEPOCEN).

In step 4, DEPOCEN contacted provided cellphone numbers, gathered consent, and interviewed consented participants. DEPOCEN collected the data and sent the data files to Vietnam Initiative. Patients did not receive any financial incentives or training for these interviews. In step 5, Vietnam Initiative analyzed the data, estimated the patient satisfaction index per participating hospital, as well as disseminated the policy reports to stakeholders and participants.

Patients or their representatives (who accompanied the patients during their hospital treatment) were followed up with shortly after being discharged (2 to 5 months) and asked about their experiences and assessments of the overall satisfaction and specific attributes of their hospital services. The telephone survey completion rates are 95% of contacted patients in 2017 and 89% of contacted patients in 2018. These response rates are deemed acceptable for data collected from a large-scale and nationwide setting. On the other side, wrong telephone numbers and inability to reach out to the provided phone numbers accounted for 40% and 55.5% of all failed contacts in 2017 and in 2018, respectively. A typical phone call lasted from 7 to 10 minutes to collect responses on 12 key questions and some demographics of patients. Finally, data from 2,927 inpatients in 2017 and 7,562 inpatients in 2018 were successfully obtained by DEPOCEN. Among these 10,489 participants, 96.7% (10,143) had responded to all survey fields necessary for this study.

## Measures

To examine the relationship between patient satisfaction levels and patients' characteristics, two sets of outcome measures were utilized in in the first analysis: (1) continuous scores on overall patient satisfaction levels and (2) three constructed binary indicators based on these

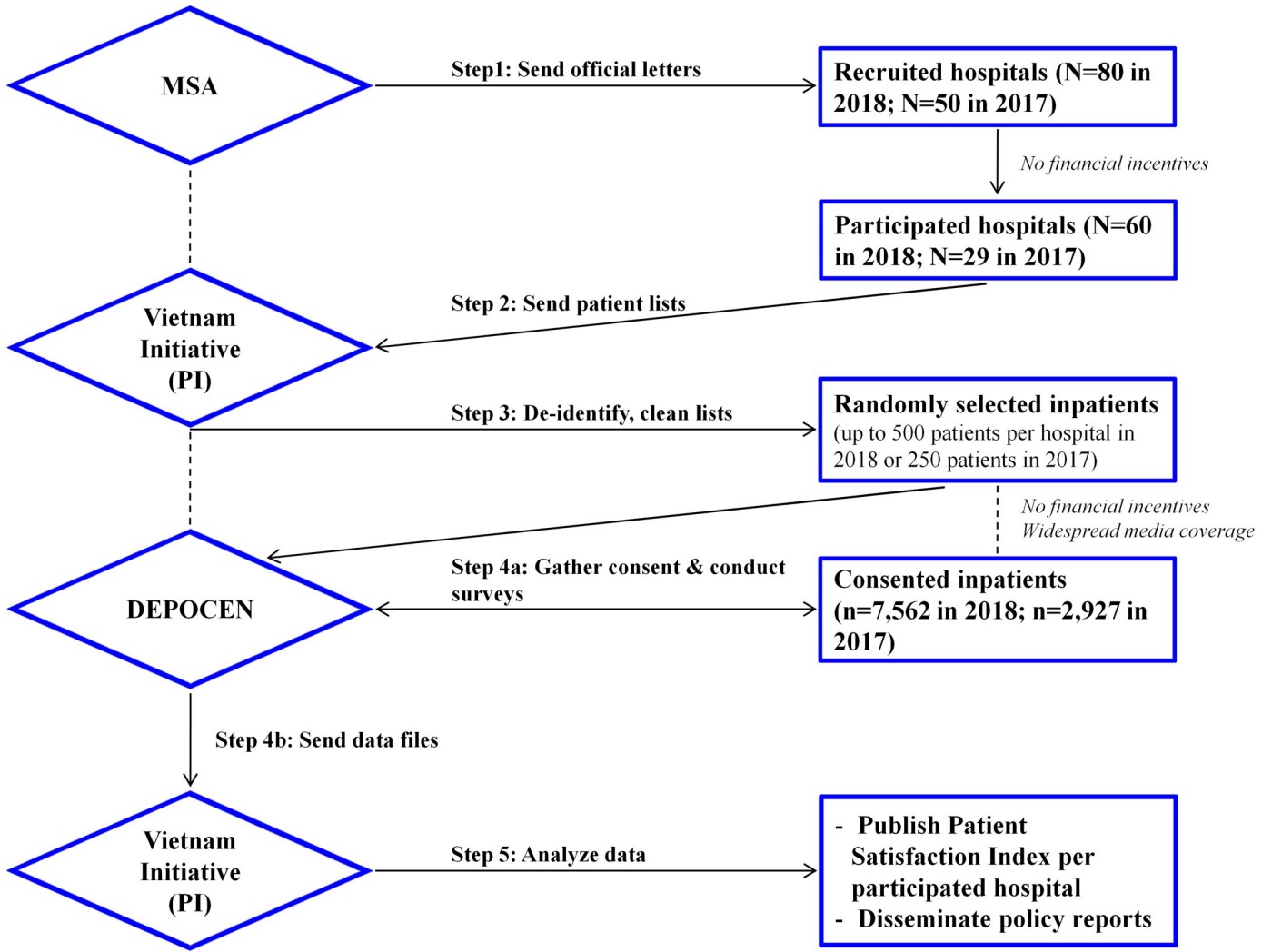

**Fig 1. Sampling approach and data collection process.**

reported scores. The patients' continuous scores reflect the degree to which patients' expectation and needs were adequately met. The second measure of overall patient satisfaction levels is a constructed binary indicator to indicate whether a patient perceived at least a threshold percent of their expectations and needs were met. We constructed three binary indicators for patient satisfaction based on patients' overall scores using three thresholds (80%, 90%, and 100%) based on the data distribution; i.e. whether a patient perceived at least 80% of their expectations and needs, at least 90% of their expectations and needs, or at least 100% of their expectations and needs. On average, 75% of patients in our sample reported at least 80% of their expectations and needs were met.

When examining patient satisfaction on three specific attributes of inpatient settings in the second analysis, we used patients' ratings on several aspects of hospital staff, facilities, and service costs. Similar to prior work on patient satisfaction [6], a 5-point scale was used for most of these questions in the surveys (except for the question of informal costs), where 1 is the lowest level of satisfaction and 5 is the highest level of satisfaction. Patient satisfaction on hospital staff consists of scores on attitudes of medical staff, access to medical staff, and medical

deliveries & instructions. Scores on hospital facilities include ratings on bed & associaries, hospital restrooms, and facility maps & directions. Patient satisfaction on treatment costs comprises scores on treatment costs, transparency in medicines & costs, and informal payments not requested.

## Statistical analysis

Using data at the patient-level, we regress each patient satisfaction outcome on: (1) specific admission characteristics, including the length of stay and extra-paid services; (2) patients' financial conditions including poverty status, employment status, and health insurance; and (3) demographic factors, such as age, gender, race, location, and level of education. For the first measure of the overall patient satisfaction level, the actual reported score, we used the Ordinary Least Square (OLS) models. Standard errors are robust to heteroskedasticity. For binary outcomes, we used logistic models with maximum likelihood estimation. For the 5-point scale ratings (ordered categorical variables) on three specific attributes of inpatient settings, we used ordered logistic regression, which preserves all the information in the 5-point scale, and avoids the need for dichotomization of patient responses.

Each regression includes hospital fixed effects, which address time-invariant unobserved factors that may differ across hospitals and communities. We also control for year fixed effects in order to capture nationwide changes. Standard errors are robust and clustered within hospitals. We performed regression analysis using Stata, version 16.0.

## Limitations

Although these hospitals were selected carefully to cover all geographical regions and administrative tiers (central, province, and district), the surveys potentially could not reflect the patient satisfaction of the entirety of Vietnam's patient populations. For instance, the insurance coverage in our data sample is 94%, which is slightly higher than the average insurance rate of Vietnam (89%). Among public hospitals, the number of involved hospitals is 69 which is fairly limited compared to the over 1,000 public hospitals in Vietnam. Furthermore, despite that private hospitals and clinics have been increasing in number in Vietnam (182 private hospitals vs. 1,183 public hospital) [17], they did not participate to the pilot surveys. Local health stations at the commune level—the third tier administrative country subdivision were not covered by the surveys of this research. These commune health stations provided 16.5% of all hospital beds in 2015 [17], however, they specialize in hygiene, vaccinations, antenatal care, and health education.

The second limitation of the current study is that all patient-reported variables were completed after discharge, creating potential measurement errors. In this context, patients were asked their experience from 2 to 5 months post treatment, which might affect the accuracy of their answers, especially for elderly patients.

## Results

### Participant characteristics

Table 1 provides basic statistics of the outcome and explanatory variables of 10,143 inpatients who had responded to all survey fields necessary for the regression analysis. The range of the observed overall patient satisfaction level is between 0% and 200%, a higher score indicates a higher level of patient satisfaction (first row). The median value and average score are 85% and 82.9%, respectively. The reported score' range is usually 0% to 100%, only 10 patients of 10,143 respondents reported a score that exceeds 100% (7 respondents with a 110% and 3 respondents with a 200% score). It implies that these patients perceived that above 100% of

**Table 1. Summary statistics.** This table reports the descriptive statistics for variables of 9,961 patients (or their relatives) with complete data from 2017 and 2018 piloted interviews in Vietnam.

| | Mean | Std. Dev | Median | Min | Max |
|---|---|---|---|---|---|
| **Measures of Patient Satisfaction Level** | | | | | |
| *Overall Patient Satisfaction Level* | | | | | |
| Percent of overall needs met [0-200] | 82.9 | (14.8) | 85 | 0 | 200 |
| Percent of patients with ≥80 of needs met [0-100] | 75.5 | (43.0) | 100 | 0 | 100 |
| Percent of patients with ≥90 of needs met [0-100] | 47.5 | (49.9) | 0 | 0 | 100 |
| Percent of patients with ≥100 of expectations [0-100] | 15.8 | (36.5) | 0 | 0 | 100 |
| *Patient Satisfaction on Hospital Staff* | | | | | |
| Attitudes of medical staff [1-5] | 4.14 | (0.81) | 4 | 1 | 5 |
| Expertise of medical staff [1-5] | 4.10 | (0.74) | 4 | 1 | 5 |
| Access to medical staff [1-5] | 4.08 | (0.73) | 4 | 1 | 5 |
| Medical deliveries & instructions [1-5] | 4.17 | (0.66) | 4 | 1 | 5 |
| *Patient Satisfaction on Hospital Facilities* | | | | | |
| Bed & associaries [1-5] | 3.92 | (0.85) | 4 | 1 | 5 |
| Hospital restrooms [1-5] | 3.69 | (0.99) | 4 | 1 | 5 |
| Facility maps/directions [1-5] | 4.06 | (0.70) | 4 | 1 | 5 |
| *Patient Satisfaction on Treatment Costs* | | | | | |
| Treatment costs [1-5] | 3.82 | (0.77) | 4 | 1 | 5 |
| Transparency in medicines & costs [1-5] | 4.01 | (0.77) | 4 | 1 | 5 |
| Informal payments not requested [0-1] | 0.91 | (0.28) | 1 | 0 | 1 |
| **Sociodemographic characteristics** | | | | | |
| Age [18-99] | 41.1 | (13.3) | 38 | 18 | 99 |
| Male [0-1] | 0.46 | (0.50) | 0 | 0 | 1 |
| Rural [0-1] | 0.54 | (0.50) | 1 | 0 | 1 |
| Race (Kinh) [0-1] | 0.92 | (0.27) | 1 | 0 | 1 |
| Postgraduate education [0-1] | 0.018 | (0.13) | 0 | 0 | 1 |
| College [0-1] | 0.14 | (0.35) | 0 | 0 | 1 |
| Vocational training [0-1] | 0.13 | (0.34) | 0 | 0 | 1 |
| High school [0-1] | 0.22 | (0.41) | 0 | 0 | 1 |
| Farmer/fisher/salt farmer [0-1] | 0.17 | (0.38) | 0 | 0 | 1 |
| Government sector [0-1] | 0.15 | (0.36) | 0 | 0 | 1 |
| Private companies/FDI [0-1] | 0.20 | (0.40) | 0 | 0 | 1 |
| Small business [0-1] | 0.12 | (0.33) | 0 | 0 | 1 |
| Hourly worker [0-1] | 0.15 | (0.36) | 0 | 0 | 1 |
| Retired, social beneficiaries [0-1] | 0.078 | (0.27) | 0 | 0 | 1 |
| Unemployed, students [0-1] | 0.12 | (0.32) | 0 | 0 | 1 |
| **Health and financial conditions** | | | | | |
| Health insurance [0-1] | 0.94 | (0.24) | 1 | 0 | 1 |
| Poor [0-1] | 0.13 | (0.34) | 0 | 0 | 1 |
| Days of hospital stay, days/stay | 10.7 | (24.1) | 7 | 0 | 1095 |
| Extra service fees [0-1] | 0.24 | (0.43) | 0 | 0 | 1 |
| Observations (patient×year) | 10,143 | | | | |

their expectation and needs were met (exceedance of complete expectation). The data on three binary measures of overall patient satisfaction indicate that 75.5 percent of all patients reported at least 80% of needs met, 47.5 percent of patients reported at least 90% of their needs met, and only 15.8 percent of all patients perceived at least 100% of their needs met.

Regarding the specific attributes of hospital services, patients are least likely to be satisfied with the hospital facilities and treatment costs. Particularly, the average score for hospital restrooms is 3.69 (where 5 is the highest level of satisfaction), the average score for bed and associaries is 3.92, and the average score for treatment costs is 3.82. A typical patient is most happy with medical staff: 4.08 for access to medical staff, 4.10 for expertise of medical staff, and 4.17 for medical deliveries & instructions.

The average age of patients was 41.1 years old and patients from rural areas accounted for 54%. These demographic and patient characteristics are relatively representative of typical Vietnam patient populations. A typical patient stayed 7 days in hospitals, which is close to the average days per inpatient (6.7 days in 2015) [17]. Patients from minority groups accounted for only 8% of all participating patients while these people consisted for 13% of the country's population [18]. Most respondents of the surveys had health insurance (94%) under their treatment, which were slightly higher than the insurance rate of Vietnam (89%). Poor patients accounted for 13% of respondents. Furthermore, 24% of patients voluntarily paid extra service fees during their hospital stays.

## Factors associated with overall patient satisfaction

Two OLS models were used to examine the association between overall patient satisfaction scores and patient characteristics. Table 2 reports linear predictors of the overall patient satisfaction score using two OLS specifications: (1) health and financial conditions only and (2) health, financial conditions, and demographics. Without controlling for sociodemographic characteristics, it was found that length of stay and poverty level of income are positively associated with the overall patient satisfaction score. For instance, a typical poor patient tends to report 1.9 percentage points higher in the overall satisfaction score than non-poor patients (p<0.01, Model 1). Insurance coverage is not a significant predictor of the overall patient satisfaction score. When controlling for rurality, employment sector, education level, gender, race, and age, these health and financial conditions are no longer significant predictors of the overall satisfaction score at the 5% level. If possessing the same values on the other predictors, rural patients tend to have 2.43 percentage points higher in satisfaction score than urban patients (p<0.01, Model 2). On average, female patients reported 1.31 percentage points higher in satisfaction score than male counterparts (p<0.01, Model 2). Patients older than 50 years of age also reported 1.79 percentage points higher than younger patients (p<0.05, Model 2). In addition, higher level of education is negatively associated with the overall satisfaction score. We also found that patients, who work in the foreign-investment and enterprise sectors, tend to have lower satisfaction levels compared to employees in government sectors, students, and social beneficiaries.

Table 3 presents the predictors of the prevalence of whether a patient perceived at least a threshold percent of whether their expectations and needs were met. We obtain similar results using this maximum likelihood estimation method, although some results are sensitive to the selected threshold. We used 80% as a baseline threshold and also reported two alternative thresholds (90% in Model 2 and 100% in Model 3). The odds ratios indicate that health insurance is positively associated with the patient satisfaction level; the odds of insured patients of being satisfied (more than 80% of their needs met) are 31% higher than non-insured patients (p<0.05, Model 1). Patients who paid extra service fees are more likely to perceive higher levels of needs fulfilled (at least 80% of their needs, by 12%, p<0.05). Patients, who had a longer length of stay (8-20 days), had a higher level of satisfaction than their peers with shorter stays. When using a higher threshold, these factors, including health insurance, extra fees, and length of stay, are no longer statistically significant (Models 2 and 3). The odds of poor patients being

**Table 2. Factors associated with overall patient satisfaction rates.** OLS Estimates on Overall Satisfaction. Standard errors are robust and clustered within hospitals.* p<0.1 ** p<0.05 *** p<0.01.

| Dep. Variable = overall patient satisfaction level | (1) | (2) |
|---|---|---|
| Health insurance [0-1] | 1.28+ | 1.22 |
| | (0.75) | (0.75) |
| Length of stay (4-7 days) [0-1] | 0.43 | 0.28 |
| | (0.41) | (0.40) |
| Length of stay (8-20 days) [0-1] | 1.51** | 1.12* |
| | (0.46) | (0.43) |
| Length of stay (≥21) [0-1] | 0.79 | 0.50 |
| | (0.65) | (0.64) |
| Extra service fees [0-1] | 0.022 | 0.67+ |
| | (0.36) | (0.36) |
| Poor [0-1] | 1.90*** | 1.17* |
| | (0.49) | (0.49) |
| Rural [0-1] | | 2.43*** |
| | | (0.49) |
| Government sector [0-1] | | 2.93*** |
| | | (0.55) |
| Small business [0-1] | | 0.67 |
| | | (0.55) |
| Farmer/fisher/salt farmer [0-1] | | 1.93*** |
| | | (0.50) |
| Hourly worker [0-1] | | 1.28* |
| | | (0.49) |
| Retired, social beneficiaries [0-1] | | 1.48+ |
| | | (0.80) |
| Unemployed, students [0-1] | | 1.80** |
| | | (0.56) |
| Postgraduate education [0-1] | | -5.15*** |
| | | (1.04) |
| College [0-1] | | -3.75*** |
| | | (0.35) |
| Vocational training [0-1] | | -1.39** |
| | | (0.42) |
| Male [0-1] | | -1.31*** |
| | | (0.36) |
| Age (≥50) [0-1] | | 1.79** |
| | | (0.53) |
| Race (Kinh) [0-1] | | -0.67 |
| | | (0.57) |
| Dep. Variable Mean | 82.93 | 82.93 |
| Dep. Variable SD | 14.83 | 14.83 |
| Observations | 10,143 | 10,143 |
| Adj R-squared | 0.07 | 0.09 |

satisfied, on two indicators of satisfaction with higher thresholds, are about 16-25% higher than their peers (two indicators statistically significant at least at the 5% level in Models 2 and 3). In all three models, the odds of a rural patient being satisfied are about 26-36% greater than for urban patients (p<0.01). In the same vein, the odds of being satisfied are positively

**Table 3. Factors associated with Likelihood to have expectations met.** The odds ratios of logistic regressions, which include hospital fixed effects and year fixed effects, were reported. Standard errors are robust and clustered within hospitals.* p<0.1 ** p<0.05 *** p<0.01.

| | (1) | (2) | (3) |
|---|---|---|---|
| **Dep. Variable = overall patient satisfaction level** | **≥80%** | **≥90%** | **≥100%** |
| Health insurance [0-1] | 1.31** | 1.06 | 1.16 |
| | (0.16) | (0.11) | (0.14) |
| Length of stay (4-7 days) [0-1] | 1.11 | 1.01 | 1.07 |
| | (0.077) | (0.066) | (0.10) |
| Length of stay (8-20 days) [0-1] | 1.18** | 1.09 | 1.07 |
| | (0.087) | (0.072) | (0.098) |
| Length of stay (≥21) [0-1] | 1.10 | 1.11 | 1.02 |
| | (0.11) | (0.11) | (0.15) |
| Extra service fees [0-1] | 1.12** | 1.03 | 1.13 |
| | (0.061) | (0.057) | (0.093) |
| Poor [0-1] | 1.17* | 1.16** | 1.25*** |
| | (0.096) | (0.076) | (0.087) |
| Rural [0-1] | 1.36*** | 1.29*** | 1.26*** |
| | (0.090) | (0.073) | (0.091) |
| Government sector [0-1] | 1.43*** | 1.39*** | 1.28** |
| | (0.12) | (0.10) | (0.16) |
| Small business [0-1] | 1.17* | 1.08 | 1.16 |
| | (0.10) | (0.088) | (0.15) |
| Farmer/fisher/salt farmer [0-1] | 1.25*** | 1.20** | 1.65*** |
| | (0.099) | (0.11) | (0.18) |
| Hourly worker [0-1] | 1.11 | 1.12 | 1.44*** |
| | (0.092) | (0.084) | (0.18) |
| Retired, social beneficiaries [0-1] | 1.26* | 1.12 | 1.54*** |
| | (0.16) | (0.12) | (0.24) |
| Unemployed, students [0-1] | 1.21** | 1.11 | 1.57*** |
| | (0.11) | (0.091) | (0.18) |
| Postgraduate education [0-1] | 0.39*** | 0.43*** | 0.40*** |
| | (0.068) | (0.088) | (0.10) |
| College [0-1] | 0.62*** | 0.56*** | 0.43*** |
| | (0.047) | (0.038) | (0.049) |
| Vocational training [0-1] | 0.81*** | 0.81*** | 0.65*** |
| | (0.057) | (0.052) | (0.062) |
| Male [0-1] | 0.80*** | 0.80*** | 0.72*** |
| | (0.045) | (0.040) | (0.051) |
| Age (≥50) [0-1] | 1.19** | 1.28*** | 1.47*** |
| | (0.092) | (0.078) | (0.13) |
| Race (Kinh) [0-1] | 0.87 | 0.86** | 0.90 |
| | (0.094) | (0.067) | (0.11) |
| Dep. Variable Mean | 0.75 | 0.48 | 0.16 |
| Dep. Variable SD | 0.43 | 0.50 | 0.37 |
| Observations | 10,143 | 10,143 | 10,143 |
| McFadden's Adj R-squared | 0.05 | 0.04 | 0.05 |

associated with lower levels of education, female gender, and older age (statistically significant at least at the 5% in all three Models).

## Perceptions on medical staff, facilities, and costs

We further examine the determinants of patient perceptions on whether or not their needs for medical practitioners were met in Table 4. The results suggest that health insurance is a positive determinant of patients' perception of their hospital staff. Particularly, insured patients are significantly more likely to express higher scores for staff expertise and treatment instructions (29% higher in the odds of reporting higher levels of satisfaction and $p < 0.05$ in Models 2; 27% higher in the odds and $p < 0.05$ in Model 4). Long-stay patients also perceived higher levels of satisfaction regarding attitudes, accessibility, and instructions than short-stay patients (Models 1, 3, and 4). For instance, the odds of reporting higher levels of satisfaction on staff attitudes for patients who stayed 8-20 days and more than 3 weeks in hospitals are 25% and 33% higher than short-stay patients (less than 4 days), respectively ($p < 0.01$ and $p < 0.05$, Model 1). In addition, patients who paid extra service fees reported to have higher levels of needs fulfilled regarding hospital staff (staff attitudes and instructions) than other patients ($p < 0.05$ in Models 1 and 4). Similar to the reaction to the overall satisfaction level, poor patients also expressed higher scores on the staff dimensions of their service experience (staff attitudes, staff expertise, and instruction). The demographic factors provide similar predictions on patient perceptions about medical staff as the reactions to the overall satisfaction level. Specifically, female patients, patients older than 50 years of age, rural patients, and those with only primary level education expressed higher scores for medical staff.

Table 5 provides odds ratios of our ordered logistic regressions on patient perceptions on hospital living arrangements and amenities. Insured patients do not statistically differ from non-insured patients when rating their experience with hospital beds, restrooms, and maps/directions. Long-stay patients (8-20 days) perceived a significantly higher level of satisfaction regarding hospital beds than their short-stay peers (22% higher in the odds of reporting a higher level of satisfaction, $p < 0.05$ in Model 1). Patients who paid extra service fees are much more likely to be satisfied with hospital beds and restrooms. In particular, the odds of these patients reporting higher levels of satisfaction with hospital beds and restrooms are 54% and 31% higher than patients who did not pay extra service fees, respectively ($p < 0.01$ in both Models 1 and 2). Overall, poor patients did not express significantly higher scores on hospital facilities when using their healthcare service in these hospitals. Similar to patient's reaction to the overall satisfaction level, patients older than 50 years of age, rural patients, and those with only primary level education expressed higher satisfaction with hospital facilities, although female patients are not more likely to report higher levels of satisfaction on hospital facilities than their male peers.

Treatment costs and transparency in stated treatment costs are important aspects of patient satisfaction. We further explore the determinants of whether a patient felt satisfied with treatment costs in Table 6. Health insurance is a strong and positive predictor on patient satisfaction on treatment costs. The odds of insured patients being satisfied with treatment costs and transparency in cost information are 57% and 32% higher than uninsured patients, respectively ($p < 0.01$ and $p < 0.05$ in Models 1-2, respectively). Poor patients did not express significantly lower scores on hospital costs. As expected, patients who paid extra service fees are less likely to be satisfied with treatment costs (11% lower in the odds of reporting high levels of satisfaction on treatment costs, $p < 0.05$ in Model 1). There is moderate evidence that these patients are more likely to be demanded to pay informal payments to hospital staff ($p < 0.10$ in Model 3). Additionally, long-stay patients (only patients with 4-7 days and more than 21 days)

**Table 4. Factors associated with patient satisfaction on hospital staff.** The odds ratios of ordered logistic regressions, which include hospital fixed effects and year fixed effects, were reported. Standard errors are robust and clustered within hospitals.* $p < 0.1$ ** $p < 0.05$ *** $p < 0.01$.

| Dep. Variable = patient satisfaction regarding | (1) Attitudes of staff | (2) Expertise of staff | (3) Access to staff | (4) Medical Instructions |
|---|---|---|---|---|
| Health insurance [0-1] | 1.11 | 1.29** | 1.14 | 1.27** |
| | (0.082) | (0.13) | (0.097) | (0.15) |
| Length of stay (4-7 days) [0-1] | 0.98 | 0.98 | 0.98 | 1.06 |
| | (0.059) | (0.052) | (0.067) | (0.067) |
| Length of stay (8-20 days) [0-1] | 1.25*** | 1.10 | 1.15* | 1.18** |
| | (0.090) | (0.067) | (0.090) | (0.088) |
| Length of stay (≥21) [0-1] | 1.33*** | 1.11 | 1.20** | 1.29*** |
| | (0.14) | (0.095) | (0.10) | (0.12) |
| Extra service fees [0-1] | 1.11** | 1.03 | 1.10* | 1.11** |
| | (0.057) | (0.056) | (0.056) | (0.060) |
| Poor [0-1] | 1.17** | 1.19*** | 1.14* | 1.18** |
| | (0.079) | (0.075) | (0.082) | (0.081) |
| Rural [0-1] | 1.17*** | 1.20*** | 1.31*** | 1.18*** |
| | (0.066) | (0.061) | (0.071) | (0.056) |
| Government sector [0-1] | 1.23*** | 1.25*** | 1.42*** | 1.25*** |
| | (0.080) | (0.066) | (0.087) | (0.081) |
| Small business [0-1] | 0.95 | 0.95 | 0.97 | 1.04 |
| | (0.072) | (0.063) | (0.067) | (0.066) |
| Farmer/fisher/salt farmer [0-1] | 1.09 | 1.23*** | 1.27*** | 1.18** |
| | (0.075) | (0.085) | (0.10) | (0.088) |
| Hourly worker [0-1] | 1.06 | 1.04 | 1.18** | 1.10 |
| | (0.071) | (0.078) | (0.084) | (0.081) |
| Retired, social beneficiaries [0-1] | 1.04 | 1.01 | 1.21* | 1.01 |
| | (0.12) | (0.11) | (0.12) | (0.11) |
| Unemployed, students [0-1] | 1.12 | 1.13 | 1.27*** | 1.10 |
| | (0.081) | (0.082) | (0.098) | (0.10) |
| Postgraduate education [0-1] | 0.69*** | 0.67*** | 0.68** | 0.68** |
| | (0.098) | (0.091) | (0.12) | (0.13) |
| College [0-1] | 0.68*** | 0.74*** | 0.71*** | 0.72*** |
| | (0.039) | (0.050) | (0.043) | (0.047) |
| Vocational training [0-1] | 0.78*** | 0.80*** | 0.76*** | 0.84*** |
| | (0.048) | (0.053) | (0.045) | (0.052) |
| Male [0-1] | 0.88*** | 0.88*** | 0.89*** | 0.94 |
| | (0.032) | (0.041) | (0.040) | (0.039) |
| Age (≥50) [0-1] | 1.44*** | 1.24*** | 1.58*** | 1.24*** |
| | (0.085) | (0.071) | (0.11) | (0.084) |
| Race (Kinh) [0-1] | 0.99 | 0.79*** | 0.93 | 0.84* |
| | (0.079) | (0.070) | (0.088) | (0.083) |
| Dep. Variable Mean | 4.14 | 4.10 | 4.08 | 4.17 |
| Dep. Variable SD | 0.81 | 0.74 | 0.73 | 0.66 |
| Observations | 10,143 | 10,143 | 10,143 | 10,143 |

are less likely to pay informal payments to hospital staff than their short-stay peers (31% lower in the odds of paying informal payments and $p < 0.01$; 38% in the odds and $p < 0.05$, respectively in Model 3). Patients older than 50 years of age are less likely to pay informal payments to hospital staff.

**Table 5. Factors associated with patient satisfaction on hospital facilities.** The odds ratios of ordered logistic regressions, which include hospital fixed effects and year fixed effects, were reported. Standard errors are robust and clustered within hospitals.* p<0.1 ** p<0.05 *** p<0.01.

| | (1) | (2) | (3) |
|---|---|---|---|
| Dep. Variable = patient satisfaction regarding | Hospital beds | Restrooms | Maps/directions |
| Health insurance [0-1] | 1.10 | 1.01 | 1.16 |
| | (0.093) | (0.073) | (0.12) |
| Length of stay (4-7 days) [0-1] | 1.05 | 0.93 | 0.93 |
| | (0.069) | (0.050) | (0.065) |
| Length of stay (8-20 days) [0-1] | 1.22** | 1.05 | 1.07 |
| | (0.089) | (0.064) | (0.076) |
| Length of stay ($\geq$21) [0-1] | 1.23+ | 1.20* | 1.12 |
| | (0.13) | (0.11) | (0.099) |
| Extra service fees [0-1] | 1.54*** | 1.31*** | 0.97 |
| | (0.12) | (0.082) | (0.051) |
| Poor [0-1] | 1.19* | 1.14* | 1.05 |
| | (0.084) | (0.071) | (0.076) |
| Rural [0-1] | 1.16** | 1.22*** | 1.22*** |
| | (0.062) | (0.069) | (0.073) |
| Government sector [0-1] | 1.37*** | 1.42*** | 1.32** |
| | (0.093) | (0.094) | (0.13) |
| Small business [0-1] | 1.02 | 1.11 | 1.12+ |
| | (0.070) | (0.076) | (0.077) |
| Farmer/fisher/salt farmer [0-1] | 1.32*** | 1.39*** | 1.32*** |
| | (0.083) | (0.098) | (0.099) |
| Hourly worker [0-1] | 1.15* | 1.19* | 1.09 |
| | (0.069) | (0.089) | (0.084) |
| Retired, social beneficiaries [0-1] | 1.08 | 1.13 | 1.22* |
| | (0.12) | (0.13) | (0.12) |
| Unemployed, students [0-1] | 1.30*** | 1.22** | 1.17* |
| | (0.095) | (0.089) | (0.092) |
| Postgraduate education [0-1] | 0.65** | 0.55*** | 0.40*** |
| | (0.10) | (0.086) | (0.092) |
| College [0-1] | 0.65*** | 0.61*** | 0.63*** |
| | (0.036) | (0.034) | (0.044) |
| Vocational training [0-1] | 0.79*** | 0.76*** | 0.90 |
| | (0.041) | (0.038) | (0.063) |
| Male [0-1] | 0.97 | 1.04 | 0.92+ |
| | (0.039) | (0.049) | (0.043) |
| Age ($\geq$50) [0-1] | 1.28*** | 1.33*** | 1.19** |
| | (0.080) | (0.073) | (0.071) |
| Race (Kinh) [0-1] | 0.92 | 1.00 | 0.96 |
| | (0.081) | (0.098) | (0.090) |
| Dep. Variable Mean | 3.92 | 3.69 | 4.06 |
| Dep. Variable SD | 0.85 | 0.99 | 0.70 |
| Observations | 10,143 | 10,143 | 10,143 |

**Table 6. Factors associated with patient satisfaction on treatment costs.** The odds ratios of ordered logistic regressions, which include hospital fixed effects and year fixed effects, were reported. Standard errors are robust and clustered within hospitals.* p<0.1 ** p<0.05 *** p<0.01.

| Dep. Variable = patient satisfaction regarding | (1) Treatment costs | (2) Transparency in medicines & costs | (3) Informal Costs requested [0-1] |
|---|---|---|---|
| Health insurance [0-1] | 1.57*** | 1.32** | 0.98 |
|  | (0.17) | (0.14) | (0.20) |
| Length of stay (4-7 days) [0-1] | 1.00 | 1.04 | 0.69*** |
|  | (0.064) | (0.066) | (0.083) |
| Length of stay (8-20 days) [0-1] | 1.14* | 1.15** | 0.83 |
|  | (0.083) | (0.078) | (0.11) |
| Length of stay ($\geq$21) [0-1] | 1.04 | 1.30*** | 0.62** |
|  | (0.096) | (0.13) | (0.12) |
| Extra service fees [0-1] | 0.89** | 1.08 | 0.79* |
|  | (0.048) | (0.058) | (0.11) |
| Poor [0-1] | 1.06 | 1.12* | 1.02 |
|  | (0.069) | (0.072) | (0.12) |
| Rural [0-1] | 0.96 | 1.17*** | 0.94 |
|  | (0.048) | (0.066) | (0.090) |
| Government sector [0-1] | 1.06 | 1.23*** | 0.96 |
|  | (0.062) | (0.067) | (0.14) |
| Small business [0-1] | 0.95 | 0.97 | 0.84 |
|  | (0.072) | (0.064) | (0.10) |
| Farmer/fisher/salt farmer [0-1] | 1.09 | 1.09 | 1.03 |
|  | (0.085) | (0.085) | (0.17) |
| Hourly worker [0-1] | 0.98 | 1.00 | 0.89 |
|  | (0.067) | (0.062) | (0.12) |
| Retired, social beneficiaries [0-1] | 0.86 | 0.93 | 0.94 |
|  | (0.097) | (0.081) | (0.19) |
| Unemployed, students [0-1] | 1.09 | 1.07 | 0.94 |
|  | (0.087) | (0.065) | (0.17) |
| Postgraduate education [0-1] | 1.06 | 0.61*** | 0.80 |
|  | (0.15) | (0.092) | (0.17) |
| College [0-1] | 0.86** | 0.69*** | 1.07 |
|  | (0.057) | (0.041) | (0.098) |
| Vocational training [0-1] | 1.02 | 0.75*** | 1.14 |
|  | (0.059) | (0.046) | (0.14) |
| Male [0-1] | 0.90** | 0.89*** | 1.13 |
|  | (0.042) | (0.037) | (0.097) |
| Age ($\geq$50) [0-1] | 0.96 | 1.17*** | 1.56*** |
|  | (0.060) | (0.067) | (0.18) |
| Race (Kinh) [0-1] | 0.79** | 0.99 | 0.95 |
|  | (0.079) | (0.066) | (0.16) |
| Dep. Variable Mean | 3.82 | 4.01 | 0.91 |
| Dep. Variable SD | 0.77 | 0.77 | 0.28 |
| Observations | 10,143 | 10,143 | 10,143 |

## Discussion

Using novel, nationwide, and large-scale patient interview data, this paper examines the associations of the patient satisfaction level and various patient characteristics, including service-specific characteristics, patients' financial conditions, and demographic factors. Compared to prior work on patient satisfaction in Vietnam, we found a higher level of patient satisfaction using a more representative sample of patients and close relatives. The average patient satisfaction score was 82.9 percent of patients' expectations and needs met. The average scores on different attributes of healthcare settings varied from 3.69/5 (lowest for hospital restrooms) to 4.17/5 (highest for medical deliveries and instruction). Using a smaller sample of patients in the North of Vietnam, Nguyen and Nguyen (2014) suggested the average satisfaction score as 3.68/5 while another study reported a lower score of 3.55/5 from a Hanoi-based group of patients [6, 7].

Our findings suggest that older patients, low-income patients, female patients, those with only primary level education, patients not working for private enterprises (or foreign enterprises), and rural patients reported higher levels of overall satisfaction. Some of these findings are contradictory to that of similar studies in Vietnam. For example, Vuong et al. (2017) found that patient satisfaction level is positively associated with patient characteristics such as younger age and higher-level of income [6]. Interestingly, our findings are consistent with some findings from prior studies in some developed countries regarding age and levels of education. For instance, two studies, one conducted in Scotland looking at 650 patients discharged from four acute care general hospitals during February and March 2002, and the second study conducted in 32 different large tertiary hospitals in the USA; both showed that patients older than 50 years of age, patients who had a shorter length of stay or better health status, and those with only primary level education reported higher satisfaction scores related to variable health service-related domains [13, 19]. Batbaatar et al. (2017) also suggested that the majority of their reviewed studies concluded that older patients were more satisfied with health services than younger patients [12]. In addition, our study suggested that female patients reported a higher level of satisfaction, which is different from the results in several studies [13, 19].

Health insurance is found to have positive influence on overall patient satisfaction (the low threshold—80 percent of expectations and needs met), primarily driven by limiting patient concerns about treatment costs, as well as increasing positive perceptions of hospital staff. This result is consistent with the finding of positive influence of insurance coverage on improving patient satisfaction in another study in Vietnam [9]. We further find that patients who paid extra fees for their hospital admission expressed higher scores for hospital living arrangements (hospital beds and restrooms) and medical staff (attitudes and instructions), but were mostly dissatisfied with treatment costs.

Similar to a study conducted in a public hospital in France [20], patients in our study reported that hospital living arrangements and amenities were their major dissatisfaction. For instance, the average score for hospital restrooms is 3.92 and the score for hospital beds is 3.92 (where 5 is the highest level of satisfaction), respectively. This exploration suggests that there is room for improvement in the public hospital environment to improve patient satisfaction and quality of care.

These findings suggest that Vietnam is achieving important steps moving forward toward universal health coverage while improving patient satisfaction. Particularly, prior to the 1980s, the government provided basic healthcare for everyone in Vietnam, until a policy shift toward a mixed market economy, leading to the inclusion of a private healthcare sector and the introduction of fee-for-service in public hospitals [21]. Vietnam has utilized a single-scheme social health insurance since 1992, first covering low-income patients and formal-sector workers

(workers in jobs with normal hours and regular pay), then extending to other populations. As of May 2019, 89% of Vietnamese populations (84 million people) were covered by this social health insurance system [22]. Private households pay for their health insurance premiums, with varying government subsidies based on the enrollee's economic status. Understanding patient satisfaction in public hospitals, the dominant healthcare clinics in Vietnam, is becoming increasingly important in this mixed public-private healthcare system. This finding suggests that Vietnam is achieving important steps moving forward toward universal health coverage while improving patient satisfaction. It is worth noting that our sample is slightly over-represented by insured patients (94% compared to the overall 89% insurance rate of Vietnam by May of 2019). Although most patients in this sample were insured, the average satisfaction score for treatment costs is 3.82, one of the lowest average scores compared to other multidimensional attributes of healthcare settings. Vietnam's policy shift toward a mixed market economy and the introduction of fee-for-service in public hospitals may play an important role in explaining this major dissatisfaction.

Patient satisfaction has been considered a mandatory barometer to evaluate how well a healthcare system is working in a number of advanced countries such as France and Germany, however, it receives limited attention in developing countries including Vietnam, where few interventions include patient satisfaction as a component. Remarkably, the most comprehensive patient surveys in Vietnam were conducted by an independent, non-governmental, non-political, and academic institution in 2017 and 2018, as the first attempt to measure patient satisfaction at a nationwide scale. Using data from these surveys, this study provides better understanding of patient satisfaction in Vietnam and identifies which sub-populations of patients and specific aspects of healthcare settings require additional attention.

## Acknowledgments

The authors would like to thank the Vietnam Initiative of Indiana University for data access. We are grateful to Jeffrey Bainter for comments. The authors also thank Livia Crim for excellent research assistance.

## Author Contributions

**Conceptualization:** Thuy Nguyen, Huong Nguyen, Anh Dang.

**Data curation:** Thuy Nguyen, Huong Nguyen, Anh Dang.

**Formal analysis:** Thuy Nguyen, Huong Nguyen, Anh Dang.

**Investigation:** Thuy Nguyen, Anh Dang.

**Methodology:** Thuy Nguyen, Anh Dang.

**Project administration:** Thuy Nguyen, Huong Nguyen.

**Writing – original draft:** Thuy Nguyen.

**Writing – review & editing:** Thuy Nguyen, Huong Nguyen, Anh Dang.

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
