## [Decision Letter · Decision Letter 0]

2 Dec 2019

PONE-D-19-21890

Determinants of Patient Satisfaction: Lessons from Large-Scaled Patient Interviews in Vietnam

PLOS ONE

Dear Dr Nguyen,

Thank you for submitting your manuscript to PLOS ONE. After careful consideration, we feel that it has merit but does not fully meet PLOS ONE’s publication criteria as it currently stands. Therefore, we invite you to submit a revised version of the manuscript that addresses the points raised during the review process.

We would appreciate receiving your revised manuscript by Jan 16 2020 11:59PM. To enhance the reproducibility of your results, we recommend that if applicable you deposit your laboratory protocols in protocols.io, where a protocol can be assigned its own identifier (DOI) such that it can be cited independently in the future. For instructions see: http://journals.plos.org/plosone/s/submission-guidelines#loc-laboratory-protocols

We look forward to receiving your revised manuscript.

Kind regards,

James M. Lightwood

Academic Editor

PLOS ONE

Additional Editor Comments:

Thank you for your submission. Both reviewers think that the manuscript is a potentially interesting and valuable contribution and worthy of publication, but also think it requires a major revision. I agree, and have requested a major revision.

Reviewer 1 presents a clear and concise list of comments and suggestions on methodological issues, and on format and presentation. I suggest each one of Reviewer 1's points be carefully addressed. Reviewer 2 agrees with some of Reviewer 1's points, and where both reviewers agree, those points should be given very careful attention. Reviewer 1 brings up an issue that escaped my notice, which is that there are no line numbers in the submission. Please ensure line numbers are in the revised submission and if you have difficulty with the PLoS ONE submission system in producing them in the revised manuscript, please contact the production staff for assistance.

Reviewer 2 raises two important concerns. The first concern is that the manuscript is vague on the primary source of the data. I interpreted the manuscript as saying that the primary data reported in the main results are from the MSA, but Reviewer 2 interprets it as saying that the data come from several sources. Please clarify this issue. Reviewer 2's second concern, based on their experience with survey research and data collection in Vietnam, is that MSA data may be biased due to influence of data collection personnel on patient responses. Please provide a careful response to Reviewer 2 on this point. If you agree that bias in the MSA data might be an important issue, I wonder if you have enough data from other sources to perform a sensitivity analysis or check on the primary source of data.

Finally, there are two important issues regarding the statistical methodology used, and for which set of data. The manuscript reports use of data that use two different scales for two different outcome variables. The first is a patient assessment of the degrees to which their expectations were met, which can exceed 100 percent; the second is a 5 point rating scale of patient satisfaction, which is converted to a 0-1 binary scale to indicated 'unsatisfied' versus 'satisfied' (Methods section page 6). However, the Results section appears to only report the second outcome measure. This issue need to be clarified. I don't see where the results on first set of outcome measures are reported. Were the measures of degree to which patient expectations were met converted to a 0-1 binary scale as well, and reported with the other type of outcome variable? Reviewer 2 seems to say that the first set of outcomes is reported, and wonders how patient satisfaction can exceed 100 percent. But I can't find where analysis for this first set of outcome is reported, and I admit I am guessing that what you mean for this set of outcomes is patient reports of the degree to which their expectations are met (so it can exceed 100 percent). Please provide more detail on the definition of this set of outcomes, and make sure it is clear to reader which set of outcomes is reported in each section of the Results.

Second, I interpret the manuscript as saying the primary results reported use logistic regression, and ordinary least squares (OLS) was used for a sensitivity analysis that is reported in a supplementary table (Results section, page 9) . However I don't see the supplementary table with the submission. If the supplementary table is to remain unpublished, then the authors should state that. If not, please ensure that any supplementary material is included in the revised submission. In any case, the results of the OLS sensitivity analysis should be described in the main text, and I can't find where it is. Reviewer 2 interprets the manuscript as reporting some main results are from the OLS regression, and says that these results should be omitted, or the analysis changed to use only logistic regression, since logistic regression is the only correct regression method. However, all the main results seem to be from logistic regression to me. So, please clarify this issue.

Regarding the issue of logistic regression versus OLS for binary data, I don't completely agree with Reviewer 2. In some cases OLS is an acceptable regression approach as well as logistic regression, and can be more robust. All the explanatory variables used for your analysis appear to be 0-1, and you have a relatively large number of observations, so some objections to OLS don't hold. The fact that all the explanatory variables are 0-1 should keep the predicted probabilities from the estimated model between 0 and 1, for example.

The question is how to compare the results from logistic and OLS regressions, since logistic regression and OLS report the coefficients on a different scale. One method is to pick a reference slope coefficient for the logistic and OLS regressions, and find the ratio of the other coefficients to the reference coefficient for the logistic and OLS regressions, respectively. The ratios should be approximately the same for each regression. The second approach is to compare the predictions of the logistic and OLS regressions. If both methods are appropriate for your data set, both should produce similar predictions, and the correctly and incorrectly classified observations should have substantial overlap. If the two methods produce substantially different predictions, the method with the best overall predictive performance should be preferred. Also, you may want to try ordered logit regression, which preserves all the information in the 5 point scale, and avoid the need for dichotomization of patient responses. Standard errors that are robust to heteroskedasticitiy should be used for the OLS regression. A routine for ordered logit regression should be provided in major statistical packages, such as SAS and Stata. These issues are discussed at length in the text by Jeffrey Wooldridge, Econometric Analysis of Cross Section and Panel Data, MIT Press, second edition, chapters 15 and 16 (I think the information is in the first edition too, but not sure in which chapters).

The authors should state what statistical package was used for the estimation (SAS, SPSS, Stata, something else?).

I also agree with both reviewers that much of the content in the manuscript is in the wrong section. For example, describing the OLS sensitivity analysis should be in the Methods section, not the Results section. Also, the descriptive statistics of the data are presented in a 'Descriptive information' section. I believe that unless there was something unexpected in the data that lead to a change in analysis, the descriptive statistics are customarily reported in the Results section.

3. Please upload a copy of Supporting Information S1 Table which you refer to in your text on page 16.

Reviewers' comments:

Reviewer's Responses to Questions

**Comments to the Author**

1. Is the manuscript technically sound, and do the data support the conclusions?

Reviewer #1: Yes

Reviewer #2: Partly

2. Has the statistical analysis been performed appropriately and rigorously? 

Reviewer #1: Yes

Reviewer #2: Yes

3. Have the authors made all data underlying the findings in their manuscript fully available?

Reviewer #1: Yes

Reviewer #2: Yes

4. Is the manuscript presented in an intelligible fashion and written in standard English?

Reviewer #1: Yes

Reviewer #2: No

5. Review Comments to the Author

Reviewer #1: General comments: This study is designed to explore determinants of Vietnamese patient satisfaction of inpatient treatment. It used a large scale of sample, that could be representative for Vietnamese population. I would recommend publishing this manuscript after revising. I have a few suggestions for improving clarity in a few places, highlighted below:

- Format:

o There is no line number as requested by the journal. It is a bit difficult to review and cite where are the issues.

o The section Conclusion should be renamed as Discussion.

- Page 3, paragraph 1: Relevant literature about correlation between patient characteristics with patient satisfaction is not sufficient, the author just focused on educational level.

- Page 5: this content is repeated at the methods section and should be in the methods section

- Page 7, 8:

o Providing a diagram could be helpful to understand the sampling approach and data.

o How long for the interview? Did participants receive any thing for their participation? What is the relationship between the research team and the surveyors? Did the surveyors receive training for interview?

o What is the software used for data analysis?

- Page 9:

o The insurance rate of Vietnam should be updated as it has been increasing significantly recent years

o Text in result should not repeat figures that mentioned in the tables

o Table 1: binary variables should be presented with percentages, not MEAN. I am a bit confused about the value of the over patient satisfaction level. Please explain how you have the overall patient satisfaction level: overall needs met with min/ max value are 0/200. This should state in the method section.

- Page 10:

o From line 19: should be presented in the discussion section

- The outcomes should be compared with similar studies in Vietnam and in other countries.

- Page 15: “Female, poor, rural patients, students, and social beneficiaries tend to have higher levels of patient satisfaction in public hospitals, the primary healthcare providers in Vietnam. These findings suggest that the public healthcare system in Vietnam still meets the medical needs for relatively vulnerable patient populations” � the language here is quite personal view, lack of evidence from this part of result to contribute for this statement. Those factors could affect to medical literacy. Patient expectations and patient satisfaction is then impacted by medical literacy level.

Reviewer #2: 15 Nov 2019

To: Authors

RE: Manuscript # PONE-D-19-21890, entitled "Determinants of Patient Satisfaction: Lessons from Large-Scaled Patient Interviews in Vietnam"

The manuscript was developed based on a recent large-scale survey in recent times, it has some merit that may be considered for publication. However, while I was reading the manuscript, I had some following comments as suggestions for improvement:

1. Introduction:

- much information referenced in this part or even in other parts of the manuscript is not updated (many cited materials were published around 2000-2004)

- in page 3 (line 10-13), I was confused about a statement "Although studies on patient satisfaction have been increasing in developing countries, publications using high-quality and large-scale patient surveys are still scarce due to resource and time constraints". When reading this, I expected that I could have an opportunity to see some discussion or supporting evidence from developing countries, yet, all of a sudden, only Turkey (developed country) and Bangladesh case studies were presented. I do believe that's not coherent and convincing as we know so many studies on patients satisfaction on healthcare were conducted and reported in both advanced and less advanced nations.

- in page 3 (line 3-11 from the bottom), instead of just listing prior studies in Vietnam, it's expected to discuss some brief results from such studies (prevalence of satisfaction, associated factors, limitations) to see the existing gaps.

- page 5 (the last paragraph), interviewing 10,489 patients is huge, I just wonder how data collectors handled this: Is it just telephone- or face-to-face-based interview or combined? More descriptions provided in methods section will help readers understand.

2. Materials and Methods:

- The info/descriptions in this part are really mixed up, it's not really clear. I suggest that authors start with describing the design and setting, then methods, data collection and analysis. It's not logical to present the procedure of regression at the beginning of this section.

- There are many pieces of information that confused readers such as pilot, why is what is pilot and why is pilot as I thought the pilot is just something the researchers did before the official study. The process of data collection is not well described such as it was said that the surveys were conducted by the Medical Services Administration (MSA) under VMOH (Vietnam Ministry of Health), Vietnam Initiative of Indiana University and Oxfam Vietnam. I think collecting data in this way could introduced a lot of bias to data. I have been working in Vietnam for more than 20 years (with focusing on research and teaching on health system, health policy and management and with a lot of interactions with policy makers in VMOH at the national and sub-national meetings and conferences; most of people raised the issue of bias about the way data were collected using the hospital system itself (such as nurses and/or doctors wearing blue/uniform clothes can not give patients freedom to report their true responses or even with data collected by MSA may not be a good way to produce valid data because MSA is just one of divisions of MOH, patients and people would be afraid of not reporting positive results in clients/satisfaction surveys because of the intervention or presence of MOH officials or staffs. Over the past years, VMOH and hospital leaders in Vietnam suggested that there be a national independent agent/organization totally independently of VMOH and the healthcare system to conduct the survey on patients satisfaction, but so far it's not feasible due to public budget limitation. In your current surveys, even you had Vietnam Initiative of Indiana Uni and Oxfarm Vietnam involved, but the main contact was still MSA of MOH.

- Also, how to have a patient's contact info is an issue. Did every single hospital asked for patients' informed consent to provide to MSA (MOH) so the surveys were conducted or without any informed consents?If this process was actually done? How was it undertaken?...

- What was the fund of the study: Authors mentioned that there was not fund for this research. However, I guessed that although there was no direct fund for incentives for patients, MOH(VMA) and/or Vietnam Initiative of Indiana Uni and Oxfarm Vietnam may actually provide some financial support for some basis logistic stuff such as contact, travel, data analysis and management, etc) because this study was really costly with alot of surveys and admin processes?

- May be there was a mistake in reporting Cronbach's alpha, as I know that this coefficient is not analysed for each item, but it is done for a group of items (say at least two items or more). If so, please correct this across the methods section as well as the whole paper.

- When authors said 95% and 89% in 2017 and 2018 participated, it's not necessary to remention "Remarkably, only 5% and 10.6% of all contacted patients refused to participate to the 2017 interviews and the 2018 interviews, respectively".

- As this study was a focus on inpatients, may be authors should use inpatients in the title because the experience and satisfaction between inpatients and outpatients are quite different

- 16% of all patients perceived over 100% of their needs met seems impossible?

3. Results:

- As this study, logistic regression was applied for inferential analysis, the OLS is not appropriate because it is only used for linear regression, while logistic regression works with the method of maximum likelihood estimation (MLE).

- As there is no discussion section, it's really difficult to distinguish between results and discussion as the authors mixed them up in one section "results'

- As this study was Vietnam-based context, it's also better to compare with other studies in Vietnam (I could not see much this comparison in this section)

- There is a quite interesting result "female patients, patients older than 50 years of age, rural patients, and those with only primary level education expressed higher scores for medical staff", please explain why or provide some possible assumptions for this as this results was just put forward in the text.

4. Conclusions:

- The first sentence was too lengthy

- The result or conclusion "Female, poor, rural patients, students, and social beneficiaries tend to have higher levels of patient satisfaction in public hospitals, the primary healthcare providers in Vietnam." could be misunderstood if no explanations are provided; perhaps these groups have low expectations and demands.

5. Limitations:

- Two other limitations are 1) cross-sectional design can not determine causality; 2) the nature of surveys conducted by MOH introduces some bias.

- Another limitation could be due to the nature of data collection conducted by MOH/MSA.

6. References:

- Not updated because just 6 among 24 items have been published within 5 recent years.

Overall, I think this manuscript might be published with major revisions required.

6. PLOS authors have the option to publish the peer review history of their article (what does this mean?). If published, this will include your full peer review and any attached files.

Reviewer #1: No

Reviewer #2: No

---

## [Author Response · Author response to Decision Letter 0]

22 Jan 2020

Please find the attached document.

---

## [Decision Letter · Decision Letter 1]

16 Jun 2020

PONE-D-19-21890R1

Determinants of Patient Satisfaction: Lessons from Large-Scaled Inpatient Interviews in Vietnam

PLOS ONE

Dear Dr. Nguyen,

Thank you for submitting your manuscript to PLOS ONE. After careful consideration, we feel that it has merit but does not fully meet PLOS ONE’s publication criteria as it currently stands. Therefore, we invite you to submit a revised version of the manuscript that addresses the points raised during the review process.

While the manuscript is much improved, Reviewer 1 is of the view that additional revisions are needed to address his/her comments. I look forward to seeing a revised version of the manuscript along with a point by point response.

We look forward to receiving your revised manuscript.

Kind regards,

David Hotchkiss

Academic Editor

PLOS ONE

Reviewers' comments:

Reviewer's Responses to Questions

**Comments to the Author**

1. If the authors have adequately addressed your comments raised in a previous round of review and you feel that this manuscript is now acceptable for publication, you may indicate that here to bypass the “Comments to the Author” section, enter your conflict of interest statement in the “Confidential to Editor” section, and submit your "Accept" recommendation.

Reviewer #1: (No Response)

2. Is the manuscript technically sound, and do the data support the conclusions?

Reviewer #1: Partly

3. Has the statistical analysis been performed appropriately and rigorously? 

Reviewer #1: No

4. Have the authors made all data underlying the findings in their manuscript fully available?

Reviewer #1: Yes

5. Is the manuscript presented in an intelligible fashion and written in standard English?

Reviewer #1: Yes

6. Review Comments to the Author

Reviewer #1: The authors tried to address all the reviewers’comments. The revision solved some of the issues that reviewers and editors pointed out and help to improve the manuscript’s quality.

However, in my opinion, this manuscript still needs more work to meet the criteria for publication. Objectives of the study were not presented clearly and concise (line 149-169). The method section was not well organized and scientific. It is very nice that the authors added a flow diagram to help visualize the sampling approach. If the authors put a bit more notice on presenting the study design, data collection and also describing major variables with their value, it would be easier for readers to follow. Such as the overall patient satisfaction level, in line 261 -265 p10, was stated that: “The score' range is usually 0% to 100%. It can exceed 100% if a patient perceived that above 100% of their expectations and needs were met (exceedance of complete expectation).” But in Table 1 p34, the range is [0-200]. Maybe I missed something…

The authors should consider about how to report the dependent variables as the previous comments and then present the results following the major objectives order. I feel confused and have to read several times to understand.

This manuscript has some merit, but I hope the authors could consider carefully about the above points for this manuscript to be ready for publication.

7. PLOS authors have the option to publish the peer review history of their article (what does this mean?). If published, this will include your full peer review and any attached files.

Reviewer #1: No

---

## [Author Response · Author response to Decision Letter 1]

8 Jul 2020

In response to comments from Reviewer 1:

R1-1. (We will number these comments consecutively in order to refer back to them as needed)

“Objectives of the study were not presented clearly and concise (line 149-169).”

Thank you. We have revised the introduction to present the objectives in a more concise fashion in lines 48-69 and lines 97-103 (the version with track changes). One example is the following: 

“This study examines factors associated with inpatient satisfaction levels in Vietnam's public hospitals during 2017-2018, using nationwide, large-scale interview data from randomized and voluntary responses to a survey provided to 69 large and public hospitals. We look at how patient satisfaction levels respond to service-specific characteristics (length of stay and extra-paid services), patients' financial conditions (employment status, health insurance, and income level), and demographic factors (age, gender, race, location and level of education). We further evaluate the associations of these factors with different salient aspects of their healthcare service experience: hospital staff, hospital facilities, and treatment costs.” 

R1-2. “The method section was not well organized and scientific. It is very nice that the authors added a flow diagram to help visualize the sampling approach. If the authors put a bit more notice on presenting the study design, data collection and also describing major variables with their value, it would be easier for readers to follow.” 

Thank you for this comment. We have made a number of edits in the method section to improve the presentations of our study design and measures. The changes are color-coded in blue (lines 187-190, 236-341, and 266-271). 

R1-3. “Such as the overall patient satisfaction level, in line 261 -265 p10, was stated that: “The score' range is usually 0% to 100%. It can exceed 100% if a patient perceived that above 100% of their expectations and needs were met (exceedance of complete expectation).” But in Table 1 p34, the range is [0-200]. Maybe I missed something…” The authors should consider about how to report the dependent variables as the previous comments and then present the results following the major objectives order. I feel confused and have to read several times to understand.

Thank you for raising this point. We made some edits to provide some details of this variable in lines 316 –323, following:

“The range of the observed overall patient satisfaction level is between 0% and 200%, a higher score indicates a higher level of patient satisfaction (first row). The median value and average score are 85% and 82.9%, respectively. The reported score' range is usually 0% to 100%, only 10 patients of 10,143 respondents reported a score that exceeds 100% (7 respondents with a 110% and 3 respondents with a 200% score). It implies that these patients perceived that above 100% of their expectation and needs were met (exceedance of complete expectation).”

---

## [Editor Report · Decision Letter 2]

4 Sep 2020

Determinants of Patient Satisfaction: Lessons from Large-Scaled Inpatient Interviews in Vietnam

PONE-D-19-21890R2

Dear Dr. Nguyen,

We’re pleased to inform you that your manuscript has been judged scientifically suitable for publication and will be formally accepted for publication once it meets all outstanding technical requirements.

Kind regards,

David Hotchkiss

Academic Editor

PLOS ONE
---

## [Editor Report · Acceptance letter]

9 Sep 2020

PONE-D-19-21890R2 

Determinants of Patient Satisfaction: Lessons from Large-Scale Inpatient Interviews in Vietnam 

Dear Dr. Nguyen:

I'm pleased to inform you that your manuscript has been deemed suitable for publication in PLOS ONE. Congratulations! Your manuscript is now with our production department. 

Kind regards, 

on behalf of

Dr. David Hotchkiss 

Academic Editor

PLOS ONE